



**Evaluation of hillslope storage with variable width under temporally varied**
**rainfall recharge**
Ping-Cheng Hsieh[1] and Tzu-Ting Huang[2]
[1] Department of Soil and Water Conservation, National Chung Hsing University, Taichung
40227, Taiwan.
[2] Department of Water Resources, Taoyuan City Government, Taoyuan 330, Taiwan.
*Corresponding author: Ping-Cheng Hsieh (pchsieh@nchu.edu.tw;
ida364@email.nchu.edu.tw)





**Abstract.** This study discussed water storage in aquifers of hillslopes under temporally varied
rainfall recharge by employing a hillslope-storage equation to simulate groundwater flow. The
hillslope width was assumed to vary exponentially to denote the following complex hillslope
types: uniform, convergent, and divergent. Both analytical and numerical solutions were acquired
for the storage equation with a recharge source. The analytical solution was obtained using an
integral transform technique. The numerical solution was obtained using a finite difference
method in which the upwind scheme was used for space derivatives and the third-order Runge–
Kutta scheme was used for time discretization. The results revealed that hillslope type
significantly influences the drains of hillslope storage. Drainage was the fastest for divergent
hillslopes and the slowest for convergent hillslopes. The results obtained from analytical solutions
require the tuning of a fitting parameter to better describe the groundwater flow. However, a gap
existed between the analytical and numerical solutions under the same scenario owing to the
different versions of the hillslope-storage equation. The study findings implied that numerical
solutions are superior to analytical solutions for the nonlinear hillslope-storage equation, whereas
the analytical solutions are better for the linearized hillslope-storage equation. The findings thus
can benefit research on and have application in soil and water conservation.
Keywords: Groundwater; Boussinesq equation; Hillslope storage; Complex hillslopes.
**1 Introduction**
Mountains in Taiwan are considerably high and steep, and the flow velocity of surface water and
subsurface water is so high that it can cause severe soil erosion on hillslopes. Therefore, the
management of catchment areas has become a crucial issue in Taiwan. Generally, hillslope form,
water transportation, sediment transport, and aquifer structure are the main factors affecting
catchment.
Some in-situ observations and experiments have investigated subsurface water flow problems.
For example, Anderson and Burt (1978) adopted an automatic system to detect soil moisture
content and found that it is significantly affected by topography. Mosley (1979) measured
overland flow and subsurface flow in a forest watershed and found that the flow discharge in a
river is greatly influenced by overland flow and subsurface flow and that the subsurface flow is
considerably decreased on mild slopes. O'Loughlin (1986) presented a topographic analysis
approach to predict the saturated zone of a watershed. McDonnell (1990) conducted an isotope
study and reported that the speed of water flow permeability in an aquifer is affected by the slope
in a watershed by means of isotope study. Genereux et al. (1993) used a chemical method to time





water flow from different upstream regions to the outlet and concluded that the travel time of flow
can be topographically determined in a watershed. Woods and Rowe (1996) also reported that
subsurface flow discharge significantly varies with topography and environmental conditions.
Subsequently, Woods et al. (1997) presented a new topographic index to predict the spatial pattern
change in subsurface flow and saturated zone thickness based on the collected data.

By contrast, some researchers have studied subsurface flow by using analytical approaches and

numerical methods. Childs (1971) first derived a generalized Boussinesq equation to delineate
groundwater flow in a sloping aquifer. Evans (1979) presented a bivariate quadrature function to
represent different topographic surfaces of catchments and further integrated terrain analysis and
slope mapping. Brutsaert (1994) linearized the Boussinesq equation and analytically solved it to
describe groundwater level. This solution provides a crucial framework to study slope features
and their hydrological response. Fan and Bras (1998) substituted Darcy's law into the continuity
equation of subsurface flow and derived an analytical solution by using the method of
characteristics. On the basis of Evans (1979), Troch et al. (2002) presented nine hillslopes to
represent the conventional hillslope types in hydrology and used the method of characteristics to
analytically solve the hillslope-storage kinematic wave equation for subsurface flow. Troch et al.
(2003) changed the variable $h$ (water depth) in the Boussinesq equation to $s$ (hill storage) and then
solved many versions of the equation by linearizing and simplifying it, using the finite difference
method to discretize the space and the multistep solver to deal with time. Later, Troch et al. (2004)
employed an exponential form to describe the variation of hillslope width and substituted it into
the linearized Boussinesq equation; then, they analytically solved the equation by using the
Laplace transform and compared the results with numerical solutions for the nonlinear hill-storage
equation with uniform rainfall recharge.

Taken together, all the aforementioned studies have indicated that geology has a considerable

influence on groundwater flow, but most studies have considered only uniform rainfall recharge
rates. Therefore, the present study employed the hill-storage Boussinesq equation of Troch et al.
(2003, 2004) to delineate groundwater flow and water storage in hillslopes but used randomly
distributed recharge rates to comply with natural rainfall recharge conditions. The present
numerical solution for the nonlinear Boussinesq equation was obtained using the finite difference
method. Discretization in space was performed using the central difference and upwind scheme,
but discretization in time was performed using the third-order total variation diminishing (TVD)
Runge–Kutta scheme. The present analytical solution to the linearized equation was acquired
using the generalized integral transforms technique presented by Özisik (1968).



## 2 Mathematical formulation

Figure 1 presents a schematic of an aquifer overlying an impermeable base with an inclined angle $\theta$. The ground surface is vegetation free, and the sole drain of groundwater is an open channel at the outlet. The aquifer was assumed to be saturated, homogeneous, and isotropic, with a constant thickness and variable width.

### 2.1 Governing equation

The continuity equation for groundwater flow with rainfall recharge yields

$$\frac{\partial s}{\partial t} = -\frac{\partial Q}{\partial x} + Rw \tag{1}$$

where $s$ is water storage [$L^2$], $Q$ is discharge [$L^3 T^{-1}$], $w$ is the hillslope width function of the flow distance $x$ [L], and $R$ is rainfall recharge [$LT^{-1}$].

Because the hillslope width in this study is not constant, the equation of hillslope width proposed by Troch et al. (2004) was introduced to delineate three hillslope types: convergent, uniform, and divergent.

$$w(x) = ce^{ax} \tag{2}$$

where $c$ is the width at the outlet [L] and $a$ is a parameter [$L^{-1}$]. The hillslope type is convergent if $a > 0$, uniform if $a = 0$, and divergent if $a < 0$.

The flow discharge obeying Darcy's law yields

$$Q = -wk_p\bar{h}(cos\theta\frac{\partial\bar{h}}{\partial x} + sin\theta) = -\frac{k_p s}{n}[cos\theta\frac{\partial}{\partial x}(\frac{s}{nw}) + sin\theta] \tag{3}$$

and then substituting Eq. (3) into Eq. (1) results in

$$\frac{\partial s}{\partial t} = \frac{k_p cos\theta}{n^2}\frac{\partial}{\partial x}[\frac{s}{w}(\frac{\partial s}{\partial x} - \frac{s}{w}\frac{\partial w}{\partial x})] + \frac{k_p}{n}sin\theta\frac{\partial s}{\partial x} + Rw \tag{4}$$

where $s \approx n \cdot \bar{h} \cdot w$, $n$ is drainable porosity, $k_p$ is hydraulic conductivity [$LT^{-1}$], and $\bar{h}$ is average water depth [L]. Note that $\bar{h}$ and $R$ are defined as

$$\bar{h} = \bar{h}(x,t) = \frac{1}{w(x)}\int_w h(x,y,t)\,dy \tag{5}$$

$$R = R(t) = \sum_{k=1}^n R_k[U(t - t_{k-1}) - U(t - t_k)] \tag{6}$$

where $R_k$ is the recharge rate within a time step and $U(-)$ is a unit step function.

Because Eq. (4) is a nonlinear equation, solving it analytically is difficult; therefore, the following linearization technique was adopted according to Troch et al. (2003):

$$\frac{s}{w} \simeq b\frac{\bar{s}_c}{\bar{w}} = bnD \tag{7}$$





where $b$ is a fitting parameter ($0 \leq b \leq 1$), $\overline{s_c}$ is average storage capacity [L$^2$], $\overline{w}$ is the
average width of the aquifer [L], and $D$ is the average aquifer depth along the hillslope.
Inserting Eqs. (2) and (7) into Eq. (4) to linearize the nonlinear term yields
$$\frac{\partial s}{\partial t} = \frac{k_p bD \cos\theta}{n} \left(\frac{\partial^2 s}{\partial x^2} - a\frac{\partial s}{\partial x}\right) + \frac{k_p}{n} \sin\theta \frac{\partial s}{\partial x} + Rce^{ax} \tag{8}$$
Equation (8) is a linearized equation and thus can be solved using an analytical approach.

### 2.2 Initial condition

The distribution of water storage was initially assumed along the $x$ direction as follows:
$$s(x,0) = \gamma n w(x) = \gamma n c e^{ax}, 0 < x < L \tag{9}$$
where $\gamma$ is the initial constant water depth [L] and $0 \leq \gamma \leq D$.

### 2.3 Boundary conditions

According to Brutsaert (1994) and Verhoest and Troch (2000), no influx occurred at the
upstream boundary condition (BC; $x = L$), that is,
$$Q = -w\overline{h}k_p\left(\cos\theta \frac{\partial \overline{h}}{\partial x} + \sin\theta\right) = 0, t > 0 \tag{10}$$
which yields
$$\frac{k_p \cos\theta}{n^2}\left[\frac{s}{w}\left(\frac{\partial s}{\partial x} - \frac{s}{w}\frac{\partial w}{\partial x}\right)\right] + \frac{k_p}{n}\sin\theta \cdot s = 0, t > 0 \tag{11}$$
Substituting Eq. (7) into Eq. (11) results in
$$\frac{k_p bD\cos\theta}{n}\frac{\partial s}{\partial x} + \left(\frac{k_p}{n}\sin\theta - \frac{ak_p bD\cos\theta}{n}\right)s = 0, t > 0, x = L \tag{12}$$
Furthermore, the outlet does not store water ($x = 0$) because water is drained out by a channel:
$$s(0,t) = 0, t > 0 \tag{13}$$

### 2.4 Analytical solution

To solve Eq. (8) associated with Eqs. (9), (12), and (13), the integral transforms presented by
Özisik (1968) were introduced as follows:
Integral transform:
$$\overline{P}(\beta_m, t) = \int_{x'=0}^{L} K(\beta_m, x') P(x', t) dx' \tag{14}$$
Inverse transform:
$$P(x,t) = \sum_{m=1}^{\infty} K(\beta_m, x) \overline{P}(\beta_m, t) \tag{15}$$
where $K(\beta_m, x)$ is the kernel function and $\overline{P}$ is the transformed function of $P$.
Before the aforementioned problem was solved, Eq. (8) was rewritten as





$\quad \frac{\partial s}{\partial t} = A\frac{\partial^2 s}{\partial x^2} + B\frac{\partial s}{\partial x} + Rce^{ax}$ (16)
$\quad$ where $A = \frac{k_p bD}{n}cos\theta, \ B = \frac{k_p}{n}sin\theta - \frac{ak_p bD}{n}cos\theta$.
$\quad$ Next, Eq. 17 is set as follows:
$\quad s(x,t) = e^{-\frac{B}{2A}x}e^{-\frac{B^2}{4A}t}s^*(x,t)$ (17)
$\quad$ and substituting Eq. (17) into Eqs. (16), (9), (12), and (13) results in
$\quad \frac{1}{A}\frac{\partial s^*}{\partial t} = \frac{\partial^2 s^*}{\partial x^2} + g(x,t)$ (18)
$\quad s^*(x,0) = \gamma nce^{\left(a+\frac{B}{2A}\right)x}, 0 < x < L$ (19)
$\quad A\frac{\partial s^*}{\partial x} + \frac{B}{2}s^* = 0, t > 0, x = L$ (20)
$\quad s^*(0,t) = 0, t > 0$ (21)
$\quad$ where $g(x,t) = \frac{R(t)c}{A}e^{(a+\frac{B}{2A})x}e^{\frac{B^2}{4A}t}$.
$\quad$ Taking the integral transform of Eqs. (18)–(21) yields
$\quad \frac{d\overline{s^*}(\beta_m,t)}{dt} + A \cdot \beta_m{}^2 \cdot \overline{s^*}(\beta_m,t) = A \cdot \bar{g}(\beta_m,t)$ (22)
$\quad \overline{s^*}(\beta_m,0) = \int_{x'=0}^{L} K(\beta_m,x')\gamma nce^{\left(a+\frac{B}{2A}\right)x'}dx' \equiv \bar{F}(\beta_m)$ (23)
$\quad$ with the kernel
$\quad K(\beta_m,x) = \left[\frac{2\left(\beta_m{}^2 + B^2/4A^2\right)}{L\left(\beta_m{}^2 + B^2/4A^2\right) + B/2A}\right]^{1/2} \cdot sin(\beta_m x)$ (24)
$\quad$ and
$\quad \bar{g}(\beta_m,t) = \int_{x'=0}^{L} K(\beta_m,x')\frac{R(t)c}{A}e^{(a+\frac{B}{2A})x'}e^{\frac{B^2}{4A}t}dx'$ (25)
$\quad$ where $\beta_m$ ($m \in \textbf{N}$, a natural number) is the root of the following eigen equation:
$\quad \beta cot(\beta L) = -\frac{B}{2A}$ (26)
$\quad$ Notably, the eigenvalue $\beta_m$ is affected by the slope $\theta$, the fitting parameter $b$, the average
$\quad$ aquifer depth $D$, and the parameter $a$.
$\quad$ Solving Eq. (22) associated with Eqs. (23)–(26) yields





$\quad \overline{s^*}(\beta_m, t) = e^{-A\beta_m^2 t}\left[\overline{F}(\beta_m) + \int_{t'=0}^{t} e^{A\beta_m^2 t'} A\overline{g}(\beta_m, t')dt'\right]$ (27)
Taking inverse transform of $\overline{s}$ results in
$\quad s^*(x, t) = \sum_{m=1}^{\infty} e^{-A\beta_m^2 t}\left[\overline{F}(\beta_m) + \int_{t'=0}^{t} e^{A\beta_m^2 t'} A\overline{g}(\beta_m, t')dt'\right]$ (28)
By employing Eq. (17), we can obtain
$\quad s(x, t) = 2ce^{\frac{-B}{2A}x}e^{\frac{-B^2}{4A}t}\sum_{m=1}^{\infty}\frac{\beta_m - \beta_m e^{\left(a+\frac{B}{2A}\right)L}\cos(\beta_m L) + \left(a+\frac{B}{2A}\right)e^{\left(a+\frac{B}{2A}\right)L}\sin(\beta_m L)}{\left(a+\frac{B}{2A}\right)^2 + \beta_m^2} \cdot$
$\quad \frac{\beta_m^2 + B^2/4A^2}{L\left(\beta_m^2 + B^2/4A^2\right) + B/2A}\left[\gamma n + R(t)\frac{e^{\left(A\beta_m^2 + \frac{B^2}{4A}\right)t}-1}{A\beta_m^2 + \frac{B^2}{4A}}\right]e^{-A\beta_m^2 t}\sin(\beta_m x)$ (29)
After the storage was obtained, the water level ($h$), discharge ($Q$), outflow rate ($q$), and relative
storage ($s_r$) were calculated, respectively, as follows:
$\quad \overline{h}(x, t) = \frac{s}{n \cdot w(x)} = \frac{2}{n \cdot e^{ax}}e^{\frac{-B}{2A}x}e^{\frac{-B^2}{4A}t}\sum_{m=1}^{\infty}\frac{\beta_m - \beta_m e^{\left(a+\frac{B}{2A}\right)L}\cos(\beta_m L) + (a+\frac{B}{2A})e^{\left(a+\frac{B}{2A}\right)L}\sin(\beta_m L)}{\left(a+\frac{B}{2A}\right)^2 + \beta_m^2} \cdot$
$\quad \left[\frac{\beta_m^2 + B^2/4A^2}{L\left(\beta_m^2 + B^2/4A^2\right) + B/2A}\right]\left[\gamma n + R(t)\frac{e^{\left(A\beta_m^2 + \frac{B^2}{4A}\right)t}-1}{A\beta_m^2 + \frac{B^2}{4A}}\right]e^{-A\beta_m^2 t}\sin(\beta_m x)$ (30)
$\quad Q(x, t) = (2B - B\cos\theta)ce^{\frac{-B}{2A}x}e^{\frac{-B^2}{4A}t}\sum_{m=1}^{\infty}\frac{\beta_m - \beta_m e^{\left(a+\frac{B}{2A}\right)L}\cos(\beta_m L) + \left(a+\frac{B}{2A}\right)e^{\left(a+\frac{B}{2A}\right)L}\sin(\beta_m L)}{\left(a+\frac{B}{2A}\right)^2 + \beta_m^2} \cdot$
$\quad \frac{\beta_m^2 + B^2/4A^2}{L\left(\beta_m^2 + B^2/4A^2\right) + B/2A}\left[e^{-A\beta_m^2 t}\gamma n + R(t)\frac{e^{\left(A\beta_m^2 + \frac{B^2}{4A}\right)t}-1}{A\beta_m^2 + \frac{B^2}{4A}}\right]\sin(\beta_m x) +$
$\quad 2Ac\cos\theta e^{\frac{-B}{2A}x}e^{\frac{-B^2}{4A}t}\sum_{m=1}^{\infty}\frac{\beta_m - \beta_m e^{\left(a+\frac{B}{2A}\right)L}\cos(\beta_m L) + \left(a+\frac{B}{2A}\right)e^{\left(a+\frac{B}{2A}\right)L}\sin(\beta_m L)}{\left(a+\frac{B}{2A}\right)^2 + \beta_m^2}$
$\quad \frac{\beta_m(\beta_m^2 + B^2/4A^2)}{L\left(\beta_m^2 + B^2/4A^2\right) + B/2A}\left[e^{-A\beta_m^2 t}\gamma n + R(t)\frac{e^{\left(A\beta_m^2 + \frac{B^2}{4A}\right)t}-1}{A\beta_m^2 + \frac{B^2}{4A}}\right]\cos(\beta_m x)$ (31)
$\quad q(t) = 2Ac\cos\theta e^{\frac{-B^2}{4A}t}\sum_{m=1}^{\infty}\frac{\beta_m - \beta_m e^{\left(a+\frac{B}{2A}\right)L}\cos(\beta_m L) + (a+\frac{B}{2A})e^{\left(a+\frac{B}{2A}\right)L}\sin(\beta_m L)}{\left(a+\frac{B}{2A}\right)^2 + \beta_m^2}$
$\quad \frac{\beta_m(\beta_m^2 + B^2/4A^2)}{L\left(\beta_m^2 + B^2/4A^2\right) + B/2A}\left[e^{-A\beta_m^2 t}\gamma n + R(t)\frac{e^{\left(A\beta_m^2 + \frac{B^2}{4A}\right)t}-1}{A\beta_m^2 + \frac{B^2}{4A}}\right]$ (32)





$s_r(x,t) = \dfrac{s}{n \cdot D \cdot w(x)} = \dfrac{2}{n \cdot D \cdot e^{ax}} e^{\frac{-B}{2A}x} e^{\frac{-B^2}{4A}t}$
$\sum_{m=1}^{\infty} \dfrac{\beta_m - \beta_m e^{\left(a+\frac{B}{2A}\right)L}\cos(\beta_m L) + \left(a+\frac{B}{2A}\right)e^{\left(a+\frac{B}{2A}\right)L}\sin(\beta_m L)}{\left(a+\frac{B}{2A}\right)^2 + \beta_m^2} \left[\dfrac{\beta_m^2 + B^2/4A^2}{L\left(\beta_m^2 + B^2/4A^2\right) + B/2A}\right] [\gamma n +$
$R(t) \dfrac{e^{\left(A\beta_m^2 + \frac{B^2}{4A}\right)t}-1}{A\beta_m^2 + \frac{B^2}{4A}}] e^{-A\beta_m^2 t}\sin(\beta_m x)$ (33)
The generalized integral transform technique was employed to acquire the above analytical
solutions because its convergence of the solution is better than that by the Laplace transform
method (Wu and Hsieh, 2019).
**2.5 Numerical method**
In addition to using an analytical approach to solve the linearized equation, Eq. (8), a
numerical model was developed to solve the original nonlinear equation, Eq. (4). With reference
to Swanson and Turke (1990), the upwind scheme and central difference of finite difference
method were used to discretize the space, and with reference to Shu and Osher (1989), the third-
order TVD Runge–Kutta scheme was applied to deal with time. The space was divided into $m$
$+ 1$ nodes with an equal interval of $\Delta x$ along the $x$ direction, in which the nodes $i = 1$ and
$i = m + 1$ are virtual outside the domain (Fig. 2). The difference equation for space
discretization of water storage is as follows:
$\dfrac{\partial s_\alpha^j}{\partial t} = \dfrac{k_p \cos\theta}{n^2}\left[\dfrac{\frac{s_\alpha^j(i+1)}{w(i+1)}+\frac{s_\alpha^j(i)}{w(i)}}{2\Delta x}\dfrac{s_\alpha^j(i+1)-s_\alpha^j(i)}{\Delta x} - \dfrac{\frac{s_\alpha^j(i)}{w(i)}+\frac{s_\alpha^j(i-1)}{w(i-1)}}{2\Delta x}\dfrac{s_\alpha^j(i)-s_\alpha^j(i-1)}{\Delta x} -\right.$
$\left.\dfrac{\left(\frac{\frac{s_\alpha^j(i+1)}{w(i+1)}+\frac{s_\alpha^j(i)}{w(i)}}{2}\right)^2}{\Delta x}\dfrac{w(i+1)-w(i)}{\Delta x} + \dfrac{\left(\frac{\frac{s_\alpha^j(i)}{w(i)}+\frac{s_\alpha^j(i-1)}{w(i-1)}}{2}\right)^2}{\Delta x}\dfrac{w(i)-w(i-1)}{\Delta x}\right] + \dfrac{k_p}{n}\sin\theta\dfrac{s_\alpha^j(i+1)-s_\alpha^j(i)}{\Delta x} + R^j w(i)$ (34)
where $i$ is the node number ($i = 1, 2, ..., m + 1$), $j$ is time, and $s_\alpha$ is the solution of order $\alpha$.
Furthermore, from Eq. (9), the initial condition becomes
$s_\alpha^j(i) = \gamma n c e^{ax_i}, \ i = 1, 2, ..., m + 1, j = 0$ (35)
and the BC becomes
1. No flux at the upstream BC
$\dfrac{1}{\Delta x}\left[\dfrac{s_\alpha^j(m+1)}{w(m+1)} - \dfrac{s_\alpha^j(m)}{w(m)}\right] + n\tan\theta = 0$
$\Rightarrow s_\alpha^j(m+1) = \dfrac{s_\alpha^j(m)}{w(m)} - n\tan\theta \cdot w(m+1) \cdot \Delta x, j > 0$ (36)
2. No storage at the downstream BC (Taylor series expansion to increase accuracy)





$\quad s_\alpha{}^j(1) = -2s_\alpha{}^j(2) + \frac{1}{3}s_\alpha{}^j(3)$ $\hfill (37)$
Regarding the discretization in time, the third-order TVD Runge–Kutta method yields the
following:
$\quad s_1 = s_0 + \Delta t \Phi(s_0)$ $\hfill (38)$
$\quad s_2 = s_1 + \frac{\Delta t}{4}[-3\Phi(s_0) + \Phi(s_1)]$ $\hfill (39)$
$\quad s_3 = s_2 + \frac{\Delta t}{12}[-\Phi(s_0) - \Phi(s_1) + 8\Phi(s_2)]$ $\hfill (40)$
where $s_1$, $s_2$, and $s_3$ are the solutions of each order. $\Phi(s) = \frac{\partial s}{\partial t}$. $s_0$ is the initial condition. The
final difference equation is listed in Appendix A.
**3 Results and Discussion**
**3.1 Verification of the analytical solution**
To validate the present analytical solution, the following parameter values presented by Troch
et al. (2004) were adopted: slope length $L = 100$ m, slope $\theta = 5\%$, $n = 0.3$, $b = 1$, $D = 2$ m, $k_p$
$= 1$ mh$^{-1}$, initial water depth $\gamma = 0.4$ m, and $R = 10$ mmd$^{-1}$. The values of the parameters in Eq.
(2) controlling the width and shape were $a = 0.02$ m$^{-1}$ and $c = 6.77$ m for a convergent
hillslope, $a = 0$ and $c = 21.627$ m for a uniform hillslope, and $a = -0.02$ m$^{-1}$ and $c =$
$50.024$ m for a divergent hillslope. The projected areas of all three hillslopes were the same
size ($2162.7$ m$^2$), so they received the same rainfall recharge. The received volume was
$4325.4$ m$^3$, and the initial water storage was also assumed to be consistent.
$\quad$ Figure 3 illustrates the spatial variation of groundwater levels for different shapes for 1, 5,
and 20 days, and Fig. 4 presents the temporal variation of flow rates at the outlet for different
shapes. Both figures reveal that our results using the generalized integral transform technique
agree well with the Laplace transform method by Troch et al. (2004), thus validating our
analytical solutions. However, we could not use the Laplace transform method directly in the
present study for two reasons. First, the inverse Laplace transform is too complex, and finding
the function is challenging even after performing the inverse Laplace transform. Second, the
convergence of solutions using our approach was better than that obtained by the Laplace
transform method as discussed in Wu and Hsieh (2019). Compared with Verhoest and Troch
(2000), whose solution summation requires the first 999 terms, namely O($10^3$), to reach
convergence, the present solution requires only the first O($10^2$) terms, leading to a convergence
that is more than 10 times faster.





**3.2 Verification of the numerical solution**
With reference to Troch et al. (2003), two cases are illustrated. Case 1 had no rainfall recharge
but did have initial water depth, and Case 2 had no initial water depth but did have rainfall
recharge. The simulated representative hillslope type was uniform ($a = 0$ and $c = 50$ m), and
the following parameters were adopted: $L = 100$ m, $\theta = 5\%$, $n = 0.3$, $D = 2$ m, $k_p = 1$ mh$^{-1}$, $\gamma$
$= 0$ and 0.4 m, and $R = 0$ and 10 mmd$^{-1}$.
Figures 5 and 6 present the variation in relative storage for Case 1 with $\gamma = 0.4$ m and Case
2 with $R = 10$ mmd$^{-1}$, respectively. Again, the results agree well with those of Troch et al. (2003),
thus validating the present numerical solution.
**3.3 Comparison between analytical solutions and numerical solutions**
With the parameters $D = 2$ m and $\gamma = 1$ m, the simulated results for convergent hillslope are
shown in Figs. 7–9, in which parameter $b$ was selected for better simulated results. When $(R, b)$
$= (50, 0.5–0.7)$, $(25, 0.3)$, and $(10, 0.2)$ in Figs. 7–9, respectively, an obvious discrepancy was
noted between the analytical and numerical solutions for different durations. The averaged
absolute relative percentage differences were 2.78% when $b = 0.2$ and 3.93% when $b = 0.7$ in
Fig. 7(a), 16.09% when $b = 0.5$ and 8.72% when $b = 0.7$ in Fig. 7(b), and 35.49% when $b = 0.5$
and 23.76% when $b = 0.7$ in Fig. 7(c). These results indicate that the discrepancy increased with
duration even when an optimal fitting parameter $b$ was selected. Similar trends can be observed
in Figs. 8 and 9. Furthermore, the shift became relatively large for a higher recharge rate ($R =$
50 mmd$^{-1}$ in Fig. 7) and smaller for a lower recharge rate ($R = 10$ mmd$^{-1}$ in Fig. 9) especially
for a longer period. Similar trends were found for a uniform hillslope when $(R, b) = (50, 0.3)$,
$(25, 0.2)$, and $(10, 0.2)$ in Figs. 10–12, respectively, and for divergent hillslope when $(R, b) =$
$(50, 0.2)$, $(25, 0.1)$, and $(10, 0.08)$ in Figs. 13–15, respectively.
Taken together, the aforementioned results imply that the present analytical solutions are
highly sensitive to the fitting parameter $b$. In fact, the parameter $b$ in Eq. (7) is affected by hill
storage, aquifer width, and aquifer thickness. Therefore, adjusting $b$ for different hillslope types
and different recharge rates can bring the analytical results closer to the numerical results. In
this study, the fitting parameter $b$ was determined using trial and error. To summarize, the
optimal parameter $b$ is relatively large for convergent hillslope but relatively small for divergent
hillslope. The parameter $b$ also increases with the recharge rate. As the recharge rate increases,
the water storage increases, and the discrepancy between both solutions also increases,
especially for convergent hillslopes.





**3.4 Variation of the remaining hill storage**
To obtain the remaining hill storage at any time for the three hillslopes with any constant
slope, the parameter $s'$, which denotes the dimensionless remaining amount of hill-storage
water, was defined as follows:
$s'(t) = \frac{\int_0^L s(x,t)dx}{\int_0^L s(x,0)dx} \approx \frac{\sum_0^{x=L} s(x,t)\Delta x}{\sum_0^{x=L} s(x,0)\Delta x}$  (41)
Because the numerical solutions were obtained by solving the nonlinear Boussinesq equation,
which is more complete, the discussion hereafter is based on the numerical model. When a 1-
day duration was used as an example with $\theta = 5\%$ and no recharge, the remaining
convergent:uniform:divergent storage ratio was approximately 1:0.984:0.888; when $\theta = 15\%$,
the ratio became 1:0.937:0.785; $\theta = 30\%$, 1:0.826:0.572; $\theta = 40\%$, 1:0.826:0.572; $\theta = 55\%$,
1:0.779:0.488; $\theta = 100\%$, 1:0.688:0.359. As expected, water drained the fastest on the steepest
divergent hillslopes. Figures 16–18 demonstrate that when the slope and simulation duration
increased, the remaining hill storage decreased. To summarize, the reduction rate of hill storage
became large for steep slopes, especially for divergent hillslopes.
**3.5 Temporally varied recharge rates effect**
Because the recharge rate is not uniformly distributed, this study considered it to have
temporal variation. Assuming $\theta = 5\%$, $D = 5$ m, and $\gamma = 0$, three patterns of recharge
distribution variation (Fig. 19) were considered to discuss their effects on hill storage. Figure
20 illustrates the spatial variation of the water table under different recharge types for a
convergent hillslope at different durations. The simulated scenarios had the same aquifer and
groundwater conditions, except for the recharge patterns. The results reveal that the water table
was significantly affected by the recharge type within a short period (12 h), but after 1 day, it
was almost no longer affected by the recharge type. Similar results were obtained for uniform
and divergent hillslopes.
We added three more recharge types: peak in the first section, peak in the last section, and
double peak; Fig. 21 illustrates the variation of outflow for different hillslopes under six types
of recharge. Figure 21(a) demonstrates that each outflow peak was different and that the
maximum peak occurred at the curve of peak in the last section, but all outflows gradually
approached one value for the convergent hillslope under the same accumulative recharge
amount. Similar results are found in Fig. 21(b) and 21(c) for uniform and divergent hillslopes,
respectively. Furthermore, the cross-sectional area at the outlet for the convergent hillslope was
relatively small; thus, the flow rate was the lowest. By contrast, the cross-sectional area at the





outlet for the divergent hillslope was relatively large; thus, the flow rate was the highest. These
hydraulic characteristics are indicated in Fig. 21(a) and 21(c). Figure 21 also illustrates that
when the recharge ceases, the outflow for convergent hillslopes decreases and then gently
increases for a long period due to the slow release of hill-stored water. For uniform hillslopes,
the outflow reduces slowly when the recharge stops, but for divergent hillslopes, the outflow
drops more rapidly owing to the fast water release.

**294   4 Concluding remarks**

To elucidate water storage of different hillslopes with variable width under any type of
temporally varied rainfall recharge, both analytical and numerical approaches were employed
to solve the Boussinesq equation. Numerical solutions to the nonlinear hillslope-storage
equation and analytical solutions to the linearized hillslope-storage equation were subsequently
presented. A summary of our findings is as follows:
1.   The analytical solutions were derived using the generalized integral transform technique

and verified with the method of Troch et al. (2004), which was derived using the Laplace

transform method. The results were consistent for convergent, uniform, and divergent

hillslopes. Our numerical solutions agreed well with the results of Troch et al. (2003),

which were obtained through the numerical integration of the partial differential equation.

2.   Although our analytical solutions were verified with previous analytical solutions, the

results need tuning of the parameter $b$ to better fit the results of the numerical model in the

same scenarios. The results reveal that as the recharge increases, $b$ increases, with $b$ being

the largest for convergent hillslopes and the smallest for divergent hillslopes.

3.   Comparison of the analytical and numerical results reveals that especially for convergent

hillslopes, when the recharge decreases, the discrepancy between the results also decreases.

4.   For the same hillslope, the hillslope storage of water decreases as the slope increases

because water drains fast along a steep slope. For the same slope and recharge distribution,

water storage is the most abundant for convergent hillslopes because of slow drainage and

least for divergent hillslopes because of rapid drainage.

The findings of the present study thus can be useful for father research and have value in the
practical application of the soil and water conservation issue.





**Appendix A**

**Difference equations of the hill-storage equation**
$s_1{}^j(i) = s_0{}^j + \Delta t \cdot \{\frac{k_p cos\theta}{n^2}[\frac{\frac{s_0{}^j(i+1)}{w(i+1)}+\frac{s_0{}^j(i)}{w(i)}}{2\Delta x}\frac{s_0{}^j(i+1)-s_0{}^j(i)}{\Delta x} - \frac{\frac{s_0{}^j(i)}{w(i)}+\frac{s_0{}^j(i-1)}{w(i-1)}}{2\Delta x}\frac{s_0{}^j(i)-s_0{}^j(i-1)}{\Delta x} -$
$\frac{(\frac{\frac{s_0{}^j(i+1)}{w(i+1)}+\frac{s_0{}^j(i)}{w(i)}}{2})^2}{\Delta x}\frac{w(i+1)-w(i)}{\Delta x} + \frac{(\frac{\frac{s_0{}^j(i)}{w(i)}+\frac{s_0{}^j(i-1)}{w(i-1)}}{2})^2}{\Delta x}\frac{w(i)-w(i-1)}{\Delta x}] + \frac{k_p}{n}sin\theta\frac{s_0{}^j(i+1)-s_0{}^j(i)}{\Delta x} + R^jw(i)\}(A.1)$
$s_2{}^j(i) = s_1{}^j + \frac{-3\Delta t}{4} \cdot \{\frac{k_p cos\theta}{n^2}[\frac{\frac{s_0{}^j(i+1)}{w(i+1)}+\frac{s_0{}^j(i)}{w(i)}}{2\Delta x}\frac{s_0{}^j(i+1)-s_0{}^j(i)}{\Delta x} - \frac{\frac{s_0{}^j(i)}{w(i)}+\frac{s_0{}^j(i-1)}{w(i-1)}}{2\Delta x}\frac{s_0{}^j(i)-s_0{}^j(i-1)}{\Delta x} -$
$\frac{(\frac{\frac{s_0{}^j(i+1)}{w(i+1)}+\frac{s_0{}^j(i)}{w(i)}}{2})^2}{\Delta x}\frac{w(i+1)-w(i)}{\Delta x} + \frac{(\frac{\frac{s_0{}^j(i)}{w(i)}+\frac{s_0{}^j(i-1)}{w(i-1)}}{2})^2}{\Delta x}\frac{w(i)-w(i-1)}{\Delta x}] + \frac{k_p}{n}sin\theta\frac{s_0{}^j(i+1)-s_0{}^j(i)}{\Delta x} + R^jw(i)\} + \frac{\Delta t}{4} \cdot$
$\{\frac{k_p cos\theta}{n^2}[\frac{\frac{s_1{}^j(i+1)}{w(i+1)}+\frac{s_1{}^j(i)}{w(i)}}{2\Delta x}\frac{s_1{}^j(i+1)-s_1{}^j(i)}{\Delta x} - \frac{\frac{s_1{}^j(i)}{w(i)}+\frac{s_1{}^j(i-1)}{w(i-1)}}{2\Delta x}\frac{s_1{}^j(i)-s_1{}^j(i-1)}{\Delta x} - \frac{(\frac{\frac{s_1{}^j(i+1)}{w(i+1)}+\frac{s_1{}^j(i)}{w(i)}}{2})^2}{\Delta x}\frac{w(i+1)-w(i)}{\Delta x} +$
$\frac{(\frac{\frac{s_1{}^j(i)}{w(i)}+\frac{s_1{}^j(i-1)}{w(i-1)}}{2})^2}{\Delta x}\frac{w(i)-w(i-1)}{\Delta x}] + \frac{k_p}{n}sin\theta\frac{s_1{}^j(i+1)-s_1{}^j(i)}{\Delta x} + R^jw(i)\}$ (A.2)
$s_3{}^j(i) = s_2{}^j + \frac{-\Delta t}{12} \cdot \{\frac{k_p cos\theta}{n^2}[\frac{\frac{s_0{}^j(i+1)}{w(i+1)}+\frac{s_0{}^j(i)}{w(i)}}{2\Delta x}\frac{s_0{}^j(i+1)-s_0{}^j(i)}{\Delta x} - \frac{\frac{s_0{}^j(i)}{w(i)}+\frac{s_0{}^j(i-1)}{w(i-1)}}{2\Delta x}\frac{s_0{}^j(i)-s_0{}^j(i-1)}{\Delta x} -$
$\frac{(\frac{\frac{s_0{}^j(i+1)}{w(i+1)}+\frac{s_0{}^j(i)}{w(i)}}{2})^2}{\Delta x}\frac{w(i+1)-w(i)}{\Delta x} + \frac{(\frac{\frac{s_0{}^j(i)}{w(i)}+\frac{s_0{}^j(i-1)}{w(i-1)}}{2})^2}{\Delta x}\frac{w(i)-w(i-1)}{\Delta x}] + \frac{k_p}{n}sin\theta\frac{s_0{}^j(i+1)-s_0{}^j(i)}{\Delta x} + R^jw(i)\} -$
$\frac{\Delta t}{12} \cdot \{\frac{k_p cos\theta}{n^2}[\frac{\frac{s_1{}^j(i+1)}{w(i+1)}+\frac{s_1{}^j(i)}{w(i)}}{2\Delta x}\frac{s_1{}^j(i+1)-s_1{}^j(i)}{\Delta x} - \frac{\frac{s_1{}^j(i)}{w(i)}+\frac{s_1{}^j(i-1)}{w(i-1)}}{2\Delta x}\frac{s_1{}^j(i)-s_1{}^j(i-1)}{\Delta x} -$
$\frac{(\frac{\frac{s_1{}^j(i+1)}{w(i+1)}+\frac{s_1{}^j(i)}{w(i)}}{2})^2}{\Delta x}\frac{w(i+1)-w(i)}{\Delta x} + \frac{(\frac{\frac{s_1{}^j(i)}{w(i)}+\frac{s_1{}^j(i-1)}{w(i-1)}}{2})^2}{\Delta x}\frac{w(i)-w(i-1)}{\Delta x}] + \frac{k}{n}sin\theta\frac{s_1{}^j(i+1)-s_1{}^j(i)}{\Delta x} + R^jw(i)\} +$
$\frac{2\Delta t}{3} \cdot \{\frac{k_p cos\theta}{n^2}[\frac{\frac{s_2{}^j(i+1)}{w(i+1)}+\frac{s_2{}^j(i)}{w(i)}}{2\Delta x}\frac{s_2{}^j(i+1)-s_2{}^j(i)}{\Delta x} - \frac{\frac{s_2{}^j(i)}{w(i)}+\frac{s_2{}^j(i-1)}{w(i-1)}}{2\Delta x}\frac{s_2{}^j(i)-s_2{}^j(i-1)}{\Delta x} -$
$\frac{(\frac{\frac{s_2{}^j(i+1)}{w(i+1)}+\frac{s_2{}^j(i)}{w(i)}}{2})^2}{\Delta x}\frac{w(i+1)-w(i)}{\Delta x} + \frac{(\frac{\frac{s_2{}^j(i)}{w(i)}+\frac{s_2{}^j(i-1)}{w(i-1)}}{2})^2}{\Delta x}\frac{w(i)-w(i-1)}{\Delta x}] + \frac{k_p}{n}sin\theta\frac{s_2{}^j(i+1)-s_2{}^j(i)}{\Delta x} + R^jw(i)$
(A.3)
**Author contribution:** Conceptualization: P.C. Hsieh; Formal analysis: T.T. Huang and P.C.
Hsieh; Funding acquisition: P.C. Hsieh; Investigation: T.T. Huang and P.C. Hsieh;



Methodology: T.T. Huang and P.C. Hsieh; Resources: P.C. Hsieh; Software: T.T. Huang;
Supervision: P.C. Hsieh; Validation: T.T. Huang; Visualization: P.C. Hsieh; Writing – original
draft preparation: P.C. Hsieh; Writing – review & editing: P.C. Hsieh.
**Competing interests:** The authors declare that they have no conflict of interest.
**Acknowledgements**
This study was financially supported by the Ministry of Science and Technology of Taiwan
under Grant No.: MOST 109-2313-B-005-037.

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



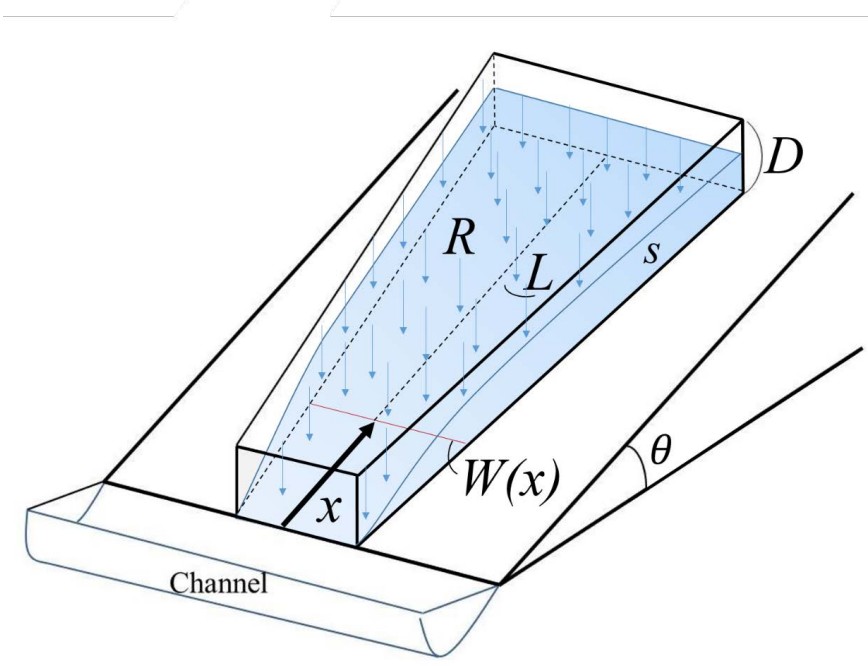

**Fig. 1.** Schematic of this study.





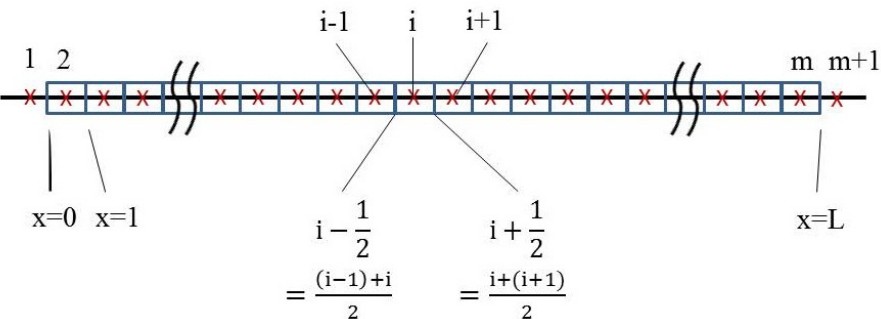

**Fig. 2.** Schematic of mesh for numerical method.





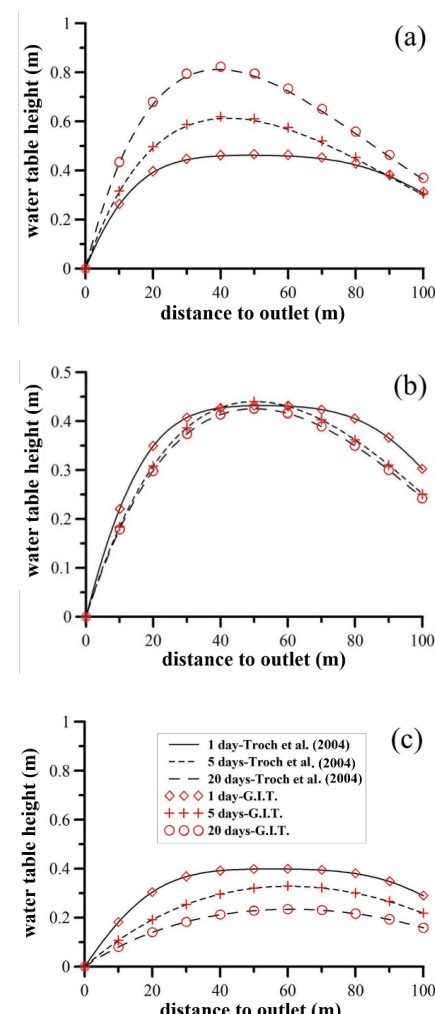

**Fig. 3.** Verification of the present solutions of groundwater levels for (a)

convergent (b) uniform, and (c) divergent hillslopes.



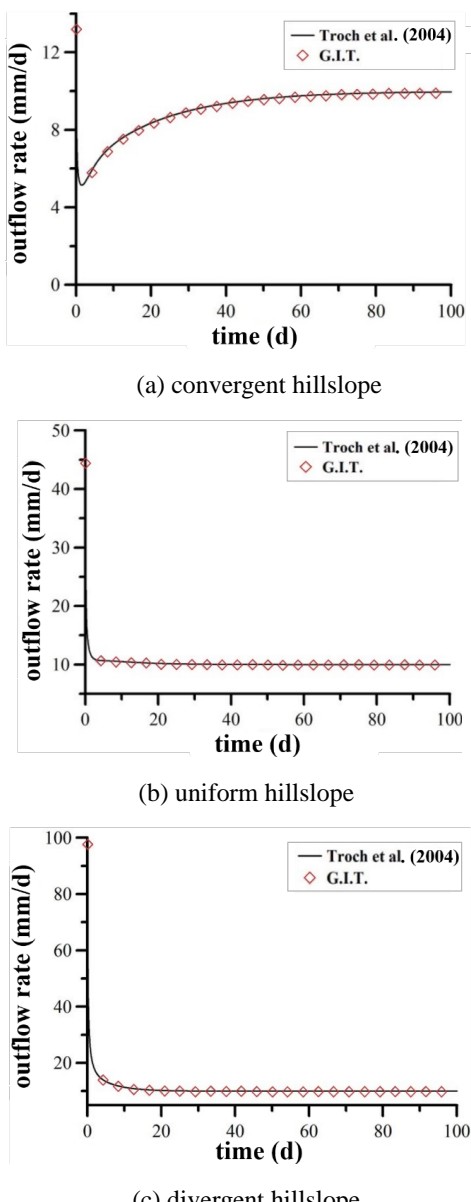

(a) convergent hillslope

(b) uniform hillslope

(c) divergent hillslope

**Fig. 4.** Verification of the present solutions of outflow hydrograph for three

hillslope types.





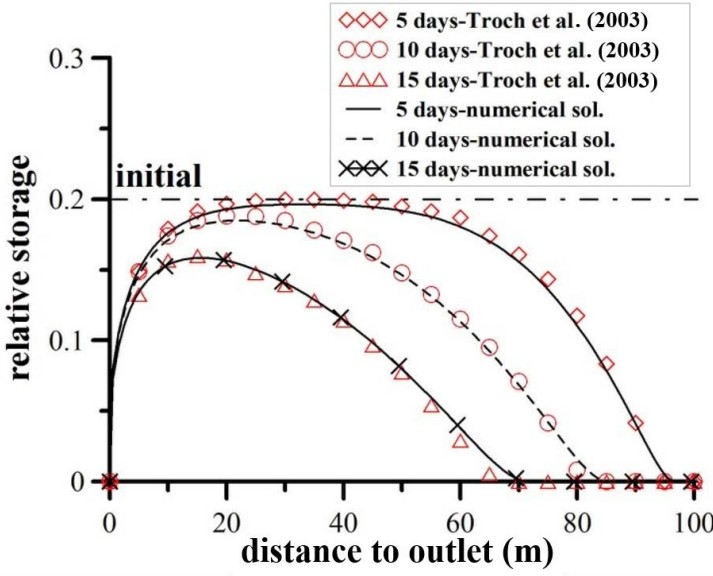

**Fig. 5.** Comparison of spatial variation of relative storage between the present

solutions and previous numerical solutions for $\gamma = 0.4$m, and $R = 0$.





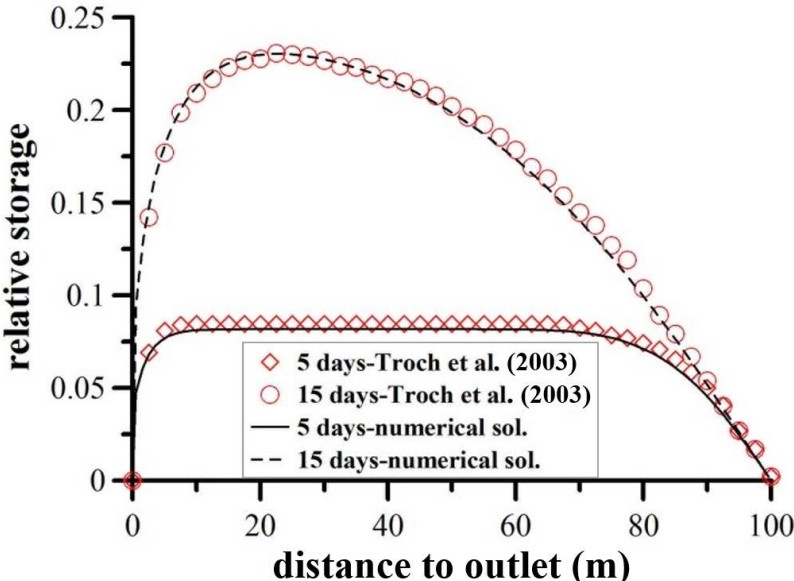

**Fig. 6.** Comparison of spatial variation of relative storage between the present
solutions and previous numerical solutions for $\gamma = 0$, and $R = 10 \mathrm{mmd}^{-1}$.

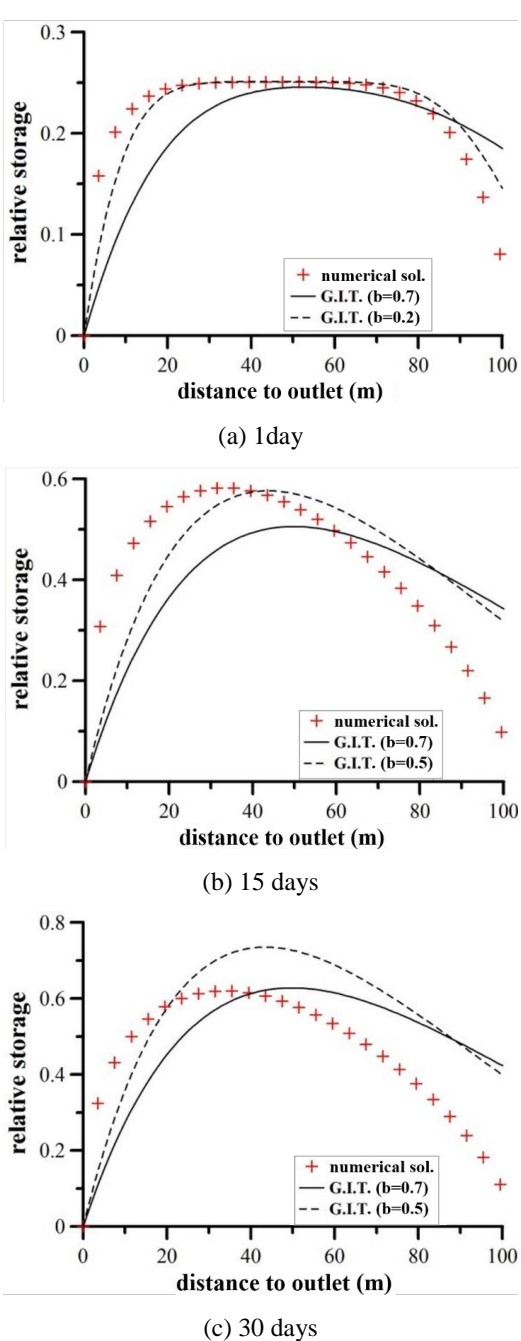

(a) 1day

(b) 15 days

(c) 30 days

**Fig. 7.** Comparison of relative storage for convergent hillslope between analytical

solutions and numerical solutions ($R = 50\,\mathrm{mm\,d^{-1}}$).





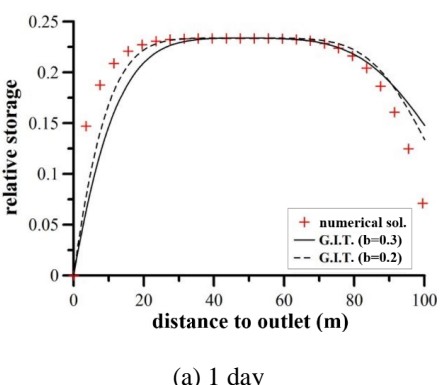

(a) 1 day

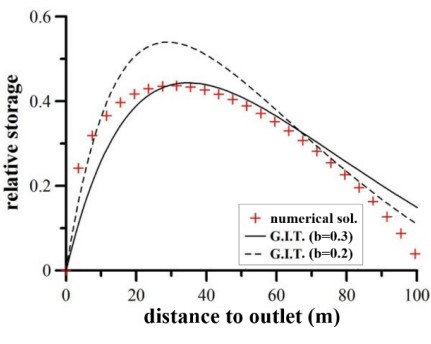

(b) 15 days

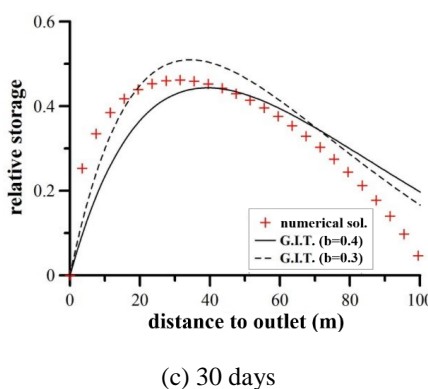

(c) 30 days

**Fig. 8.** Comparison of relative storage for convergent hillslope between analytical

solutions and numerical solutions ($R = 25\,\text{mmd}^{-1}$).



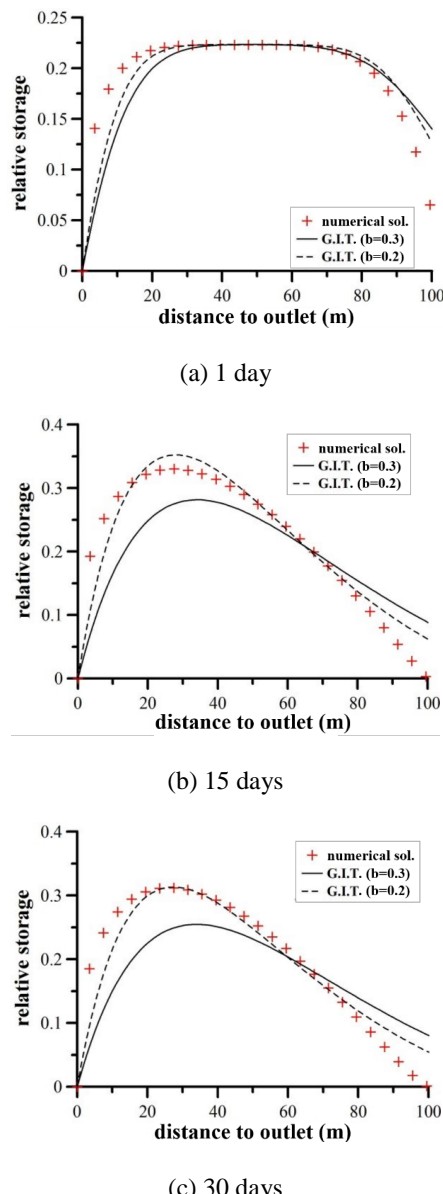

(a) 1 day

(b) 15 days

(c) 30 days

**Fig. 9.** Comparison of relative storage for convergent hillslope between analytical

solutions and numerical solutions ($R = 10\text{mmd}^{-1}$).



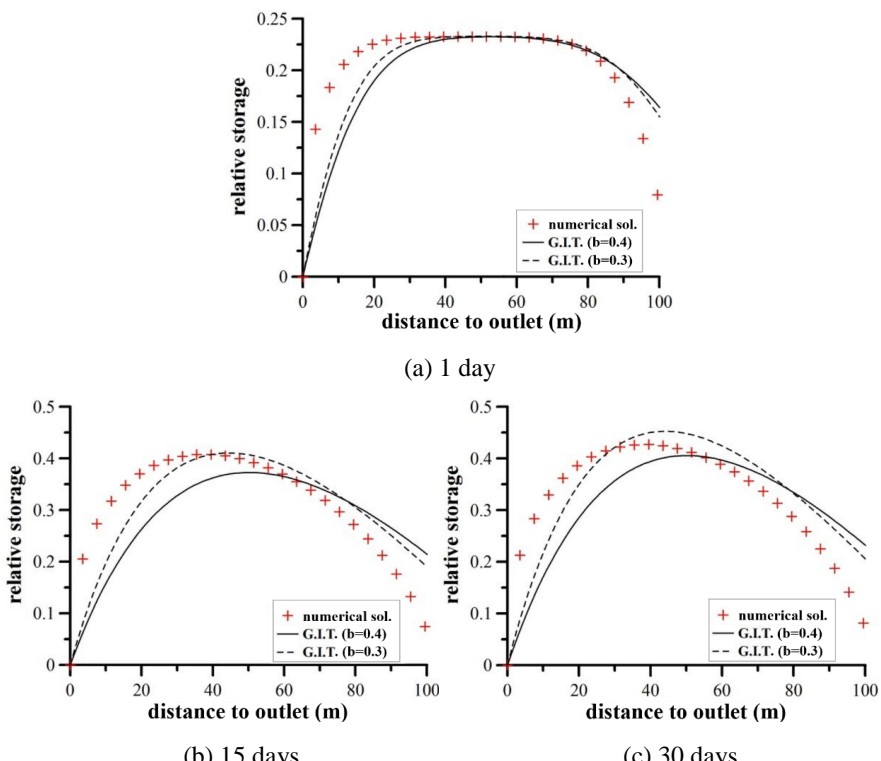

(a) 1 day

(b) 15 days          (c) 30 days

**Fig. 10.** Comparison of relative storage for uniform hillslope between analytical solutions and numerical solutions ($R$=50mmd$^{-1}$, $\theta$=5%).

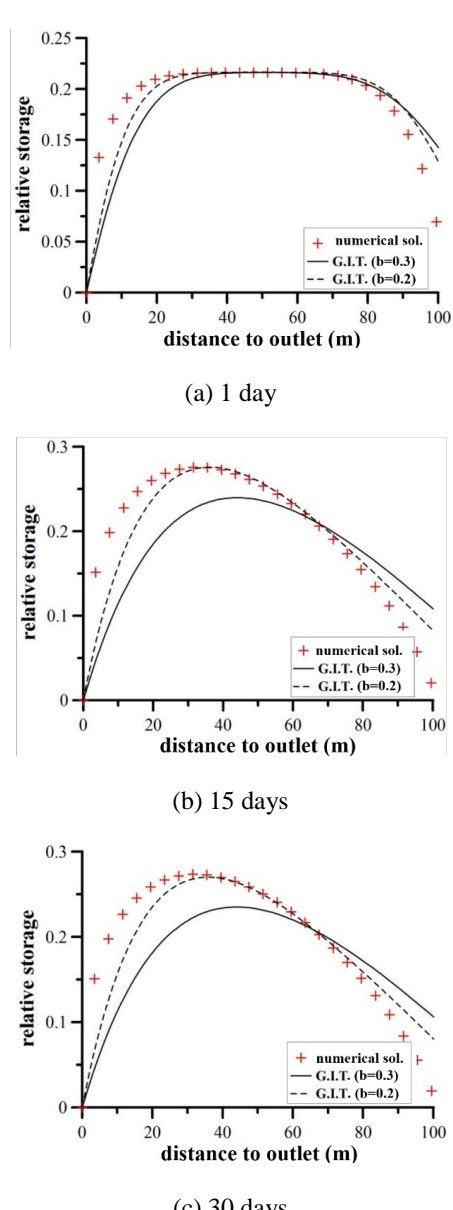

(a) 1 day

(b) 15 days

(c) 30 days

**Fig. 11.** Comparison of relative storage for uniform hillslope between analytical solutions and numerical solutions ($R$=25mmd$^{-1}$, $\theta$=5%).



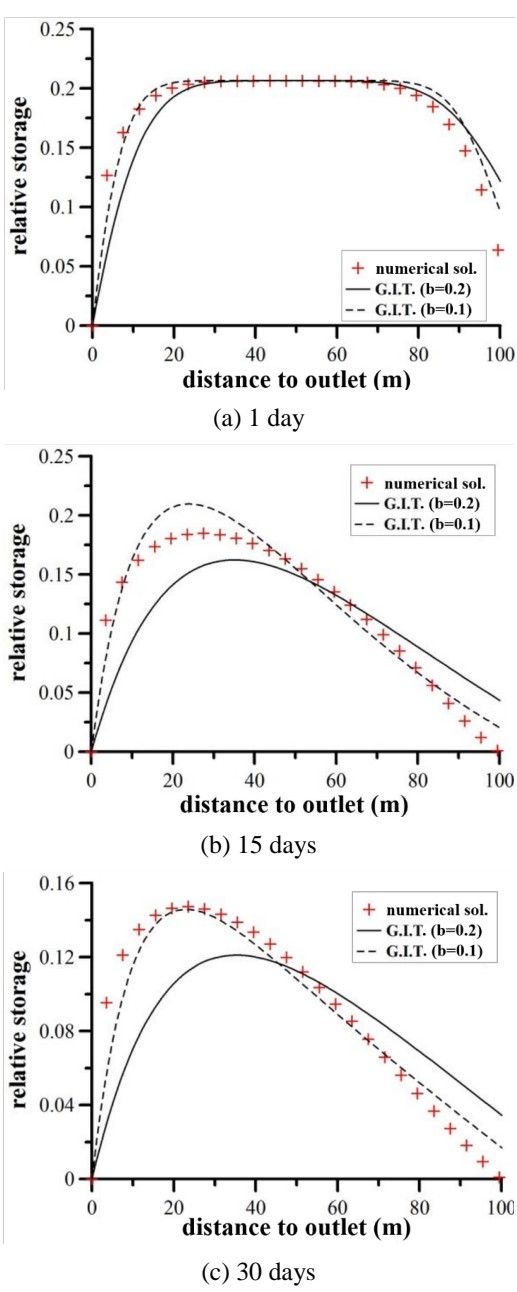

(a) 1 day

(b) 15 days

(c) 30 days

**Fig. 12.** Comparison of relative storage for uniform hillslope between analytical solutions and numerical solutions ($R$=10mmd$^{-1}$, $\theta$=5%).



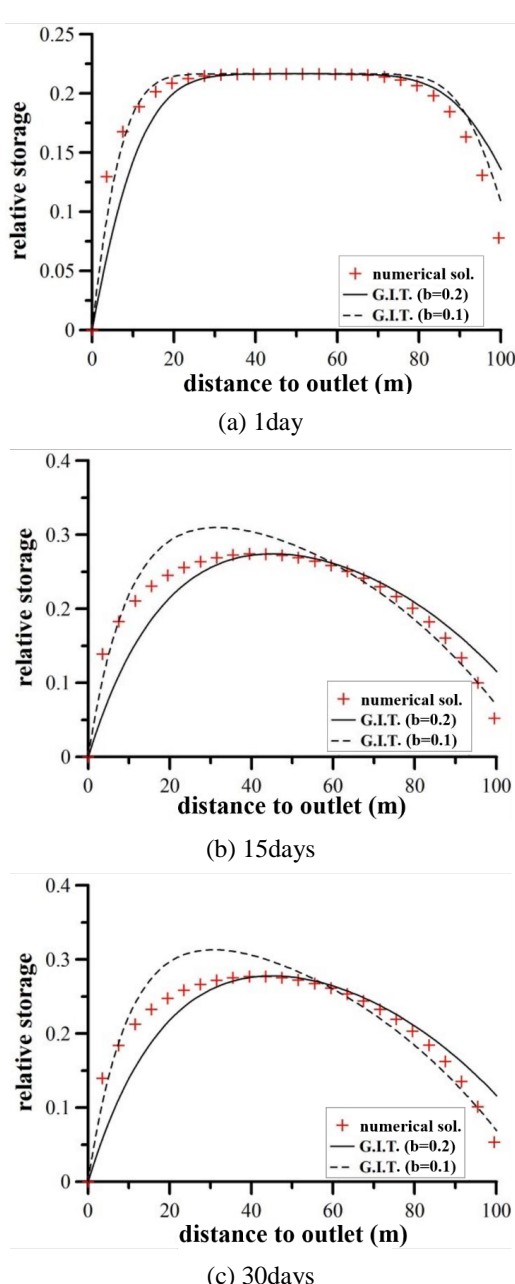

(a) 1day

(b) 15days

(c) 30days

**Fig. 13.** Comparison of relative storage for divergent hillslope between analytical solutions and numerical solutions ($R$=50mmd$^{-1}$, $\theta$=5%).

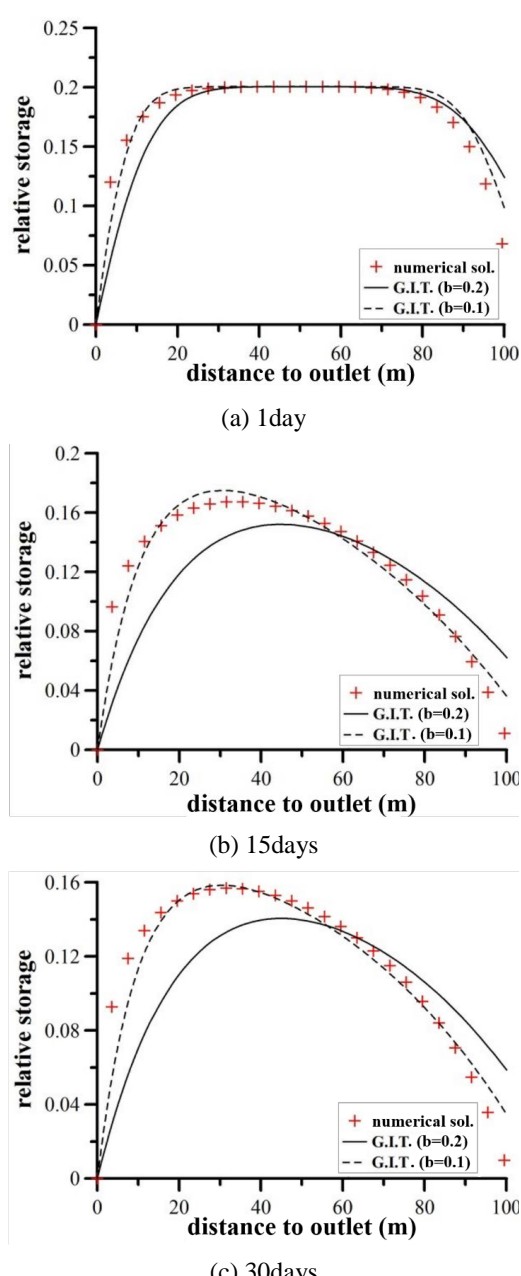

(a) 1day

(b) 15days

(c) 30days

**Fig. 14.** Comparison of relative storage for divergent hillslope between analytical solutions and numerical solutions ($R$=25mmd$^{-1}$, $\theta$=5%).



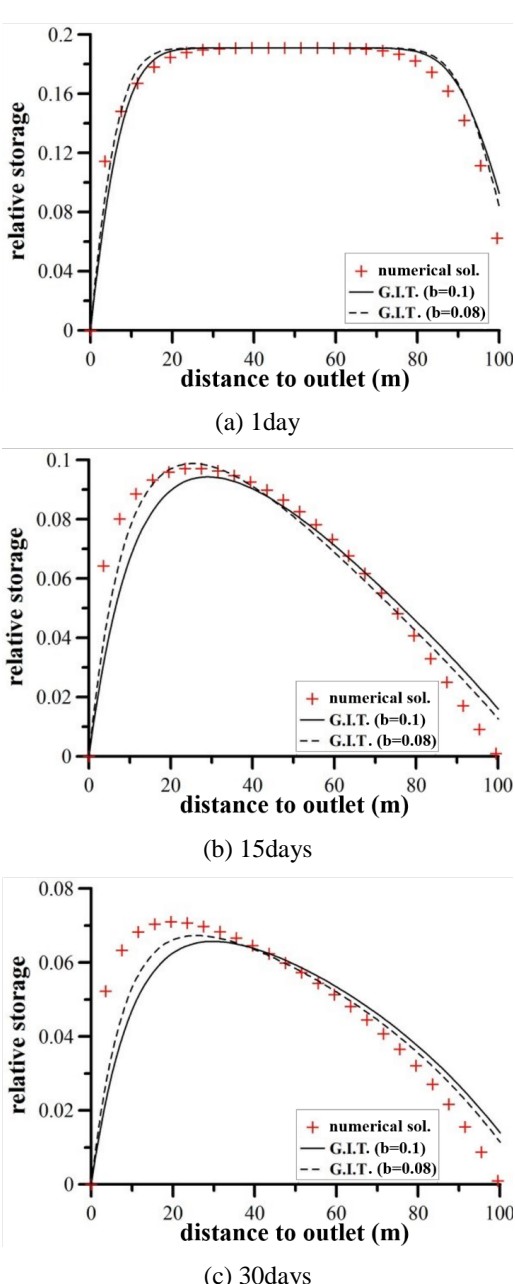

(a) 1day

(b) 15days

(c) 30days

**Fig. 15.** Comparison of relative storage for divergent hillslope between analytical solutions and numerical solutions ($R$=10mmd$^{-1}$, $\theta$=5%).



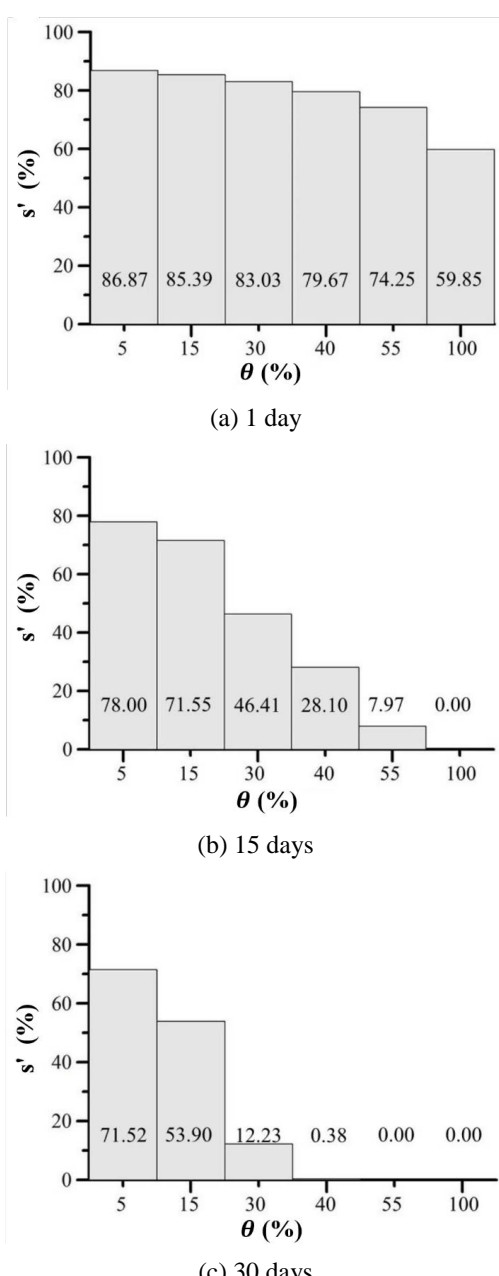

(a) 1 day

(b) 15 days

(c) 30 days

**Fig. 16.** Variation of the ratio of storage to initial storage at different durations for

convergent hillslope.



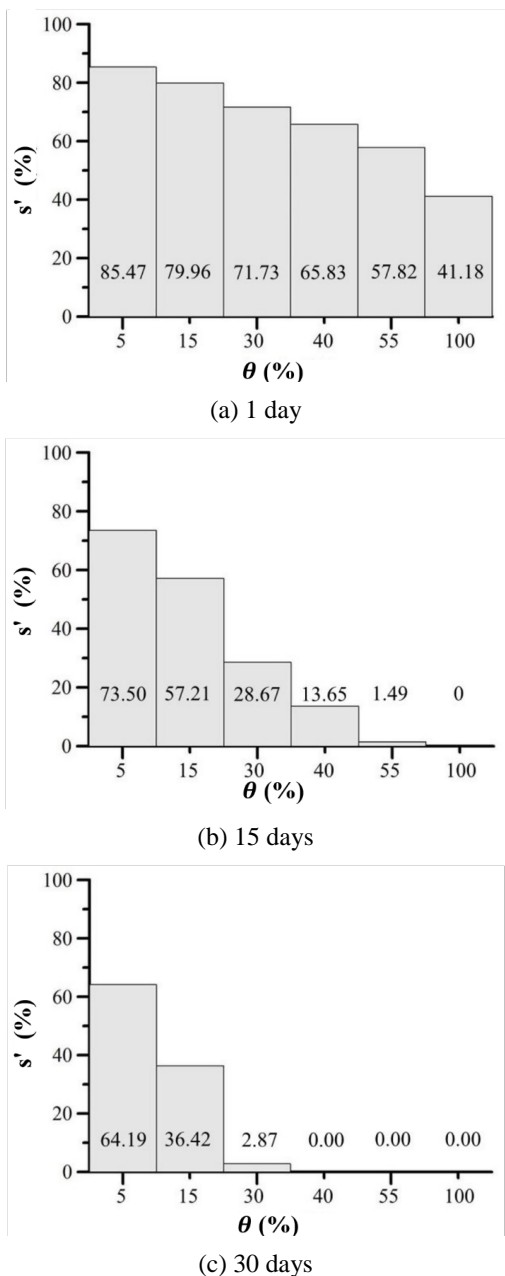

**Fig. 17.** Variation of the ratio of storage to initial storage at different durations for

uniform hillslope.





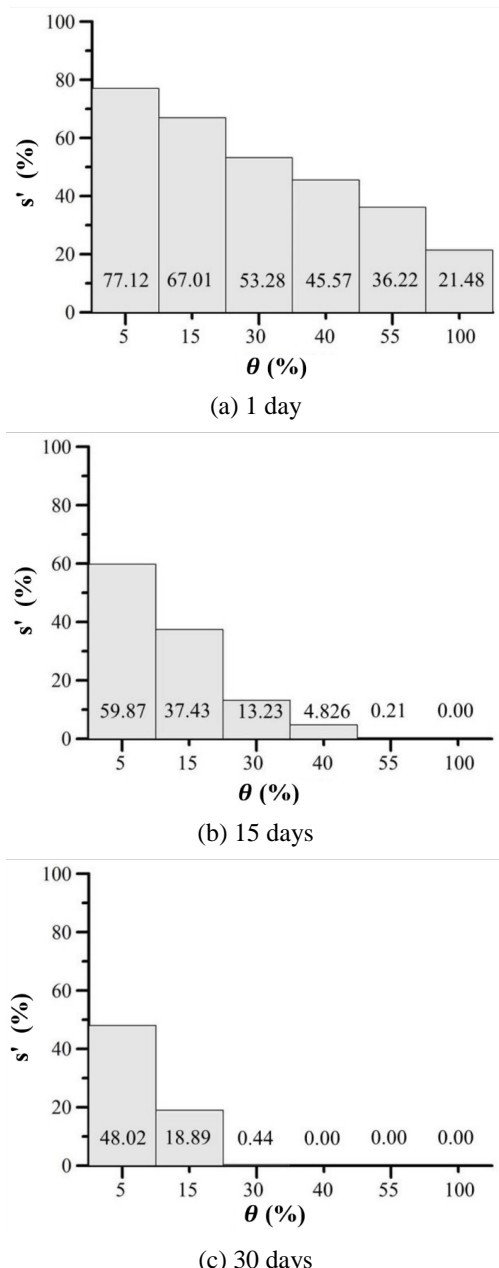

**Fig. 18.** Variation of the ratio of storage to initial storage at different durations for

divergent hillslope.





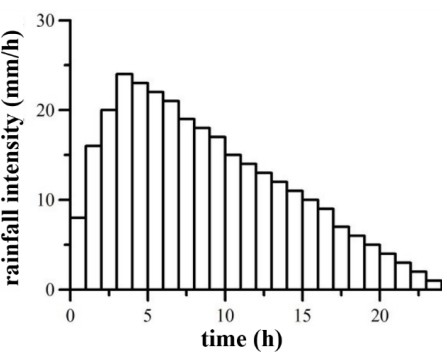

(a) peak at the first quarter section

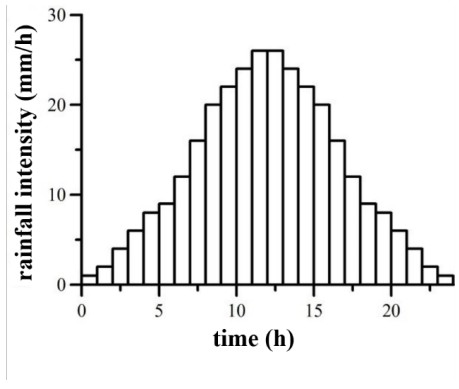

(b) peak at center

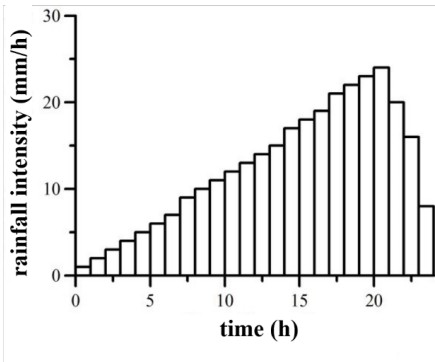

(c) peak at the third quarter section

**Fig. 19.** Presumed patterns of temporally various distributed recharge rates.





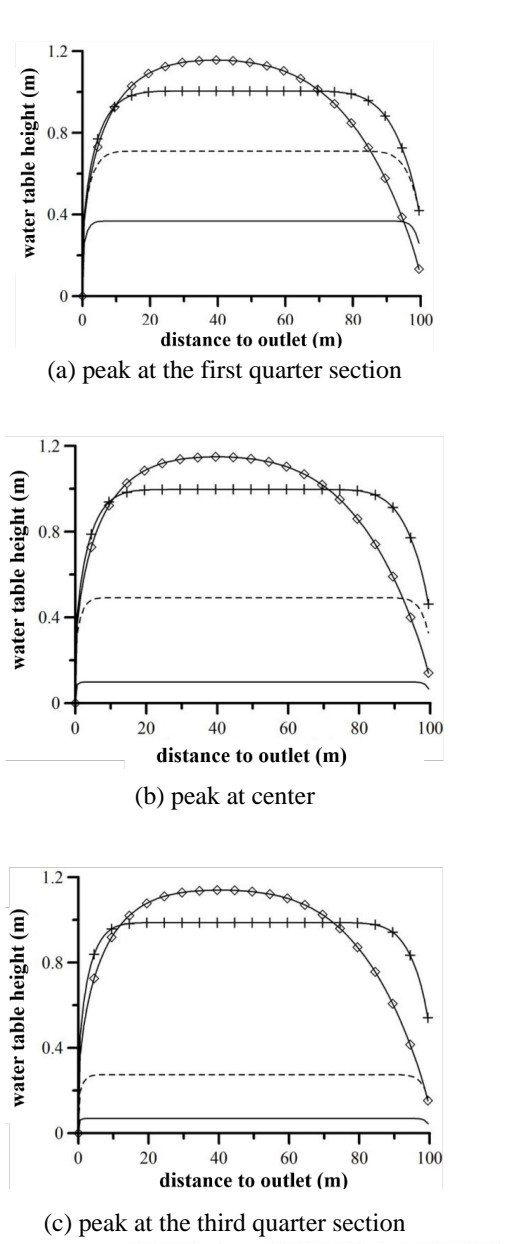

Fig. 20. Variation of water table for three patterns of recharge distribution for
convergent hillslope.





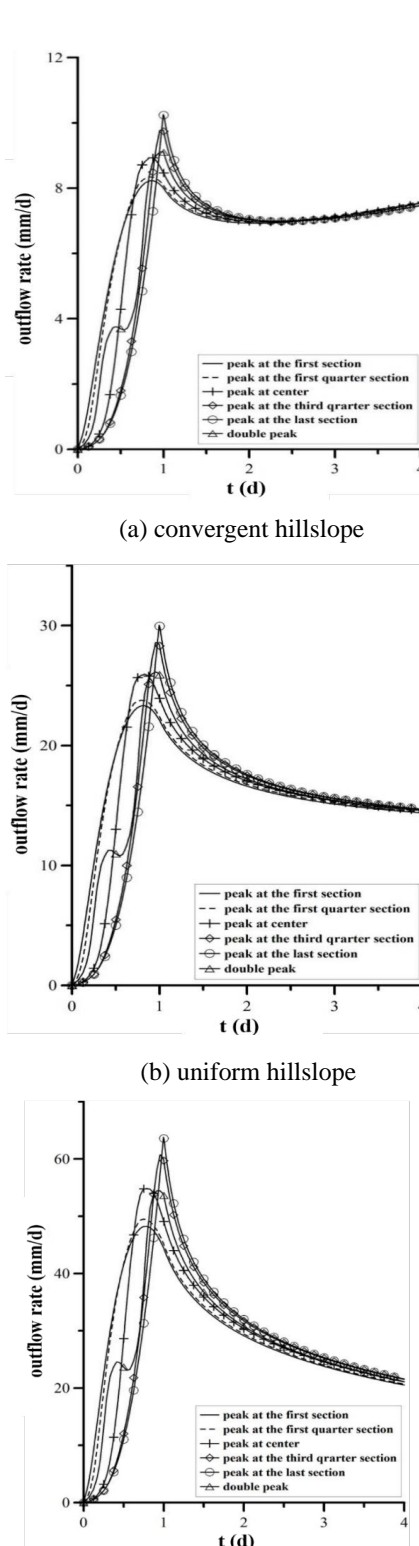

(a) convergent hillslope

(b) uniform hillslope

(c) divergent hillslope

**Fig. 21.** Hydrograph of outflow for three different hillslopes under six types of recharge distribution.