# Peer review of "Evaluation of hillslope storage with variable width under temporally varied"

_Hydrology and Earth System Sciences, 2021_

## Author Comment (AC1)

In this manuscript, a new analytical solution to the Boussinesq equation for variable widths and recharge rates is presented and analyzed.

I am in favor of the idea of the paper, and the paper is reasonable well written. However, there are a number of issues:

- Huyck et al. (2005) presented an analytical solution to the Boussinesq equation (in a different form) for variable widths and recharge rates. This is highly relevant work and has not been discussed. For example, equation 6 implies that the recharge is constant within each time step, which is exactly the same approach as Huyck et al. (2005).

Thank you very much for your comment. After examining the study of Huyck et al. (2005), we could find that in their study the Boussinesq eq. is in a different form, and they derived the analytical solutions by the Laplace transform method for different time steps and then took a summation of all the solutions. The derivation process of analytical solutions is clear but a little complicated when compared with ours. Although the concept of our Eq. (6), meaning that the recharge is constant within each time step, is the same as Huyck et al. (2005), the expressions are different. Don't you think our expression is more concise and neater? We will add the citation and discussion of Huyck et al. (2005) to the article. Thank you again.

- It should be clarified how the results from Troch et al. (2004) were obtained. Were these provided by any of the authors of that paper? Lines 216 and further indicate that the authors did not code these analytical solutions. Then how were these results obtained.

Thank you very much for your comment. The results obtained from Troch et al. (2004) were not provided by any of the authors of that paper. Because the analytical solutions are given as Eqs. (33) and (35) of the paper of Troch et al. (2004), anyone can use them to reproduce the results in the paper. We will revise the description as follows:

"Both figures reveal that our results using the generalized integral transform technique agree well with the analytical solutions derived by the Laplace transform method, i.e. Eqs. (33) and (35) in Troch et al. (2004), thus validating our analytical solutions."

- The statement on line 222 is problematic: Verhoest and Troch (2000) do not state anywhere that they require 999 terms. They only state that after so many terms the residuals become insignificant, but they never performed an analysis on this. Usually, with these solutions, the results become stable after less than 100 summations.

Thank you very much for your comment. We mistook the meaning "summation of the first 999 terms" from Verhoest and Troch (2000), and we will revise the description as follows:

  "As stated in Verhoest and Troch (2000), after solution summation of the first 999 terms, namely $O(10^3)$, the residuals become insignificant. In the present study, the solution summation of the first $O(10^2)$ terms, usually less than 50, could reach convergence. The convergence of the present solution is better."

- My major concern is the statement on line 247: the analytical solution is supposed to be highly sensitive to the fitting parameter b. When comparing a numerical solution to an analytical solution, the results should ALWAYS be equal, regardless of the parameters that are used. The only exception is when oscillations are obtained, but then either the temporal or spatial discretization should be modified. Looking at figures 8 through 15, it is clear that the discrepancies are too large, and something must be wrong. I did not check the mathematical solution, but either there is an issue there, and/or there is something wrong in the coding, and/or the numerical solution has issues. This is something that must be corrected before the paper can be accepted.

Thank you very much for your comment. The reviewer said "When comparing a numerical solution to an analytical solution, the results should ALWAYS be equal, regardless of the parameters that are used." I totally agree with this point when both solutions are derived from the same governing equations, initial/boundary conditions and input parameters. However, an analytical solution to a LINEARIZED governing equation is possibly not equal to a numerical solution to a NONLINEAR governing equation. This present analytical solution is obtained for a linearized equation, but the present numerical solution is for a nonlinear equation which was described on Line 177 (original version of MS) "a numerical model was developed to solve the original nonlinear equation, Eq. (4)". Both solutions are to different governing equations, so

there are discrepancies in between. For the numerical solution by a finite difference method (F.D.M.) to the same LINEARIZED equation, the results are given below:

[Figure]

It shows that the numerical solutions are equal to the analytical solutions based on the same governing equations and same scenarios, thus justifying that the present analytical solutions are correct.

- Line 304 states that the results from Troch et al. (2003) were obtained by solving their equation numerically. Line 230-231 states that the numerical solutions of Troch et al. (2003) matches the newly developed numerical solution

well. This supports my suspicion that something is not right with the new analytical solution.

Thank you very much for your comment. The present numerical solutions are for the nonlinear governing equation in our study. In Troch et al. (2003), they derived a numerical solution by finite difference for the same nonlinear equation, Equation (6) in their study. Both results match each other, and this justifies the present numerical solutions in our study are correct. The present analytical solutions to the linearized equation have been justified correct as shown in the response of last comment.

- There are too many figures in the paper. Something like 12 figures for a paper of this length should be the maximum. For example, I do not think that figure 2 is needed. The comparison with Troch et al. should be presented in less figures, as well as the comparison between the numerical and analytical solutions.

Thank you very much for your comment. Original Figures 2, 5, 8, 11, and 14 are deleted now.

**Thank you very much for all of your precious comments and suggestions.**

---

## Author Comment (AC2)

Comment on hess-2021-50

This study applied both analytical and numerical approaches to solve hillslope hydrological dynamics equation, and tested (as well as compared) the results in some idealized situations. However, the manuscript was written more like a mathematic article though dealing with a practical problem in hydrology. Thus I think some major revisions are needed to meet the criteria of HESS. Please see my detailed comments as following.

Major comments:

Just as I mentioned, simulating the outlet discharge water of a hillslope is a practical hydrological problem. While many mathematic tools can be employed to solve the problem, the only metric to evaluate them is to compare their outcomes with some real observed measurements. However, this study stopped by testing its framework in some idealized conditions without checking with the real situation and data. On the other side, the topic of explicitly solving hillslope hydrology is not new. In fact, based on my knowledge, some land models have already employed the conception of hillslope and solve its hydrology dynamics explicitly using numerical solutions. These models have been tested and applied at different scales, and the observations are also available at different scales. So at such stage, conducting a similar research but only in idealized conditions is not decent for publication in HESS (maybe more suitable for a journal for applied mathematics). To overcome this shortage, the authors may consider using some real data to configure and evaluate their model, even at a local scale. Thus it can let us see more clearly the ability of each (analytical or numerical) methods and benefit future research. Please note that all required real data must be available as hydrological modelers have already depicted and validated the hillslope from local to global scales. So I see no excuse to refuse this suggestion.

Response: Thank you for your precious comments, and I quite respect your opinion. Analytical approaches and numerical methods are commonly used two ways to do research. Analytical solutions are focused on their logic processes and systematic derivation. The limitation of analytical solutions is that the geometry (or shape) of simulated domain must be specific and regular. Conversely, numerical models can be applied to irregular geometry in a wide range. In the past, the reviewer usually asks us to compare our new analytical solutions with numerical solutions to validate the correctness of our analytical solutions. Nowadays, the analytical solutions have been compared with numerical solutions developed by ourselves as well as other analytical solutions. The reviewer presented that the present analytical solutions as well as

numerical methods need to test by the real situation and data. We totally agree that an analytical solution could be verified by real data, but this is not the only way. As you know, both analytical and numerical solutions cannot conform with the real observation data because of the high uncertainty and irregular properties of soils, which influence the input parameters, in a real aquifer. Some input parameters in real situation are hard to be determined or acquired, for example, real hydraulic conductivity, real porosity, and real inclined angle of slope in a watershed. These parameters vary with space in a real watershed. There are no available data in the published papers listed in the references section. We searched for available real data and input parameters after the reviewer gave the comments. The following papers were found:

Matonse A. H. and Kroll C. (2009). "Simulating low streamflows with hillslope storage models". water resources research, 45, W01407, doi:10.1029/2007WR006529, 2009.

In the above paper, the parameters $a$ and $c$ of the hillslope equation in Eq. (2), $w(x) = ce^{ax}$, are lacked because they didn't give $w(x)$. Moreover, the hydraulic conductivity and soil porosity only give their lower limits and upper limits.

Kong, J, Shen C., Luo Z., Hua G., and Zhao H. (2016) "Improvement of the hillslope-storage Boussinesq model by considering lateral flow in the unsaturated zone". water resources research, 52, 2965–2984, doi:10.1002/ 2015WR018054.

In the above paper, the authors studied saturated and unsaturated soil layers including sand, loam, and clay. The shape and wide are not the same with ours, i.e. not $w(x) = ce^{ax}$. Porosity varies with space and time. No real data is given for verification.

Norbiato D, Borga M. (2008) "Analysis of hysteretic behaviour of a hillslope-storage kinematic wave model for subsurface flow". Advance in Water Resources 31, 118-131.
In the above paper, the thickness and porosity of the aquifer are not given. No real data is given for verification.

Based on the above literature surveying, we can find there is no sufficient information for us to compare the present solutions with real data/situation.

If we conduct the observation and measurement of the real data and parameters, we shall need a huge financial support and spent a lot of time. Therefore, we used different ways to verify our analytical solutions and numerical solutions in this study.

The analytical solutions were compared with the published paper of Troch et al. (2004), and the numerical solutions were compared with Troch et al. (2003). The comparison results are in a good agreement, and thus which validates both the solutions. In addition, we also made a comparison between the numerical model and the analytical solutions to check the correctness of the analytical solutions, and conversely, the analytical solutions can also benefit the verification of the numerical model. To develop a more popular numerical model associated with the map of GIS is our goal for future research.

Furthermore, this present analytical solution is obtained for a linearized equation, but the present numerical solution is for a nonlinear equation which was described on Line 177 (original version of MS) "a numerical model was developed to solve the original nonlinear equation, Eq. (4)". Both solutions are to different governing equations, so there are discrepancies in between. For the numerical solution by a finite difference method (F.D.M.) to the same LINEARIZED equation, the results are given below:

[Figure]

[Figure]

It shows that the numerical solutions are equal to the analytical solutions based on the same linearized governing equations and same scenarios, thus justifying that the present analytical solutions are correct.

We have tried our best to do the research, and prepared the present manuscript for a long journey. Thank you very much for your precious comments.

Specific comments:

L41, "by means of isotope study": Please delete these words.

Response: Thank you for your comment. These words have been deleted. Please see Line 41.

L77, "The ground surface is vegetation free, …": Please discuss the potential effects of vegetation.

Response: Thank you for your comment. The vegetation effect on the hill-storage is a good topic. The potential effects of vegetation might be beneficial to the water storage. Please allow us to do this in future research. The present study only discusses vegetation free surface.

L98, Equation (6): The n here should not be mixed with the n for drainable porosity.

Response: Thank you for your comment. We will change the upper limit of summation n to M. Please see Line 101.

L102, Equation (7): s/w=nh=bnD, because b<1, so h<D? But D is the average depth, how can h be less than its average everywhere?

Response: Thank you for your comments. In a real world, the groundwater h might be greater than the average depth D, but in this study h<D is the limitation. In other words, only pore water storage was considered.

L103, "where b is a fitting parameter …": Please show more detail for the method used in tuning b.

Response: Thank you for your comment. We will add the following explanation" which is determined by trial and error to better fit the results of the numerical model." Please see Lines 106-107.

L194, Equation (37): Please show more detail how to use Taylor series expansion to transform the Eq (13) to the Eq(37).

Response: Thank you for your comment. We have added more detail about using Taylor series expansion above Eq. (37). Please see Lines 198-201.

From linear extrapolation, we have

$$
\begin{cases}
s_\alpha{}^j(1) = s_\alpha{}^j(0) - \dfrac{\Delta x}{2} s_\alpha{}^{j\prime}(0) + \dfrac{1}{2!}\left(\dfrac{\Delta x}{2}\right)^2 s_\alpha{}^{j\prime\prime}(0) + O(\Delta x)^3 \\[2mm]
s_\alpha{}^j(2) = s_\alpha{}^j(0) + \dfrac{\Delta x}{2} s_\alpha{}^{j\prime}(0) + \dfrac{1}{2!}\left(\dfrac{\Delta x}{2}\right)^2 s_\alpha{}^{j\prime\prime}(0) + O(\Delta x)^3 \\[2mm]
s_\alpha{}^j(3) = s_\alpha{}^j(0) + \dfrac{3\Delta x}{2} s_\alpha{}^{j\prime}(0) + \dfrac{1}{2!}\left(\dfrac{3\Delta x}{2}\right)^2 s_\alpha{}^{j\prime\prime}(0) + O(\Delta x)^3
\end{cases}
$$

Eliminating $s_\alpha{}^{j\prime}(0)$ and $s_\alpha{}^{j\prime\prime}(0)$ with boundary condition $s_\alpha{}^j(0) = 0$, we can obtain

$$
s_\alpha{}^j(1) = -2s_\alpha{}^j(2) + \tfrac{1}{3} s_\alpha{}^j(3)
$$

L203-232: What is the major difference between this work and Torch et al. (2003, 2004)? The authors should particularly stress it in the manuscript because the similarity is too high in my view based on the current description.

Response: Thank you for your comment. As mentioned in the introduction section, Troch et al. (2003) solved the linearized and simplified Boussinesq equation using the finite difference method to discretize the space and the multistep solver to deal with time. Troch et al. (2004) analytically solved the linearized Boussinesq equation with uniform rainfall recharge by using the Laplace transform. However, in this study, we

evaluate the hillslope storage with variable width under temporally varied rainfall recharge, not uniform recharge. This is more realistic in real situation. Thus, the source term R=R(t) (see Eq. (6)) in the governing equation makes it difficult to be solved by an analytical approach. We used the generalized integral transform instead of Laplace transform to avoid the difficulty which inverse Laplace transform might encounter. This is a new contribution. We will stress it in the conclusion section. Please see Lines 311-315.

L263: "Theta = 5%": Is the theta angle of slope? How to understand the symbol of percentage?

Response: Thank you for your comment. Yes, theta is the angle of slope. Theta = 5% means Theta = 0.05 in radian. Please see Line 269.